# Few-Shot Text Classification with Dual Contrastive Consistency Training

## Abstract

In this paper, we explore how to utilize pre-trained language model to perform few-shot text classification where only a few annotated examples are given for each class. Since using traditional cross-entropy loss to fine-tune language model under this scenario causes serious overfitting and leads to sub-optimal generalization of model, we adopt supervised contrastive learning on few labeled data and consistency-regularization on vast unlabeled data. Moreover, we propose a novel contrastive consistency to further boost model performance and refine sentence representation. After conducting extensive experiments on four datasets, we demonstrate that our model (FTCC) can outperform state-of-the-art methods and has better robustness.

## 1 Introduction

Text classification is a fundamental task in natural language processing with various applications such as question answering (Rajpurkar et al., 2016), spam detection (Shahariar et al., 2019) and sentiment analysis (Chong et al., 2014). With the advancement of deep learning, fine-tuning pre-trained language model(Devlin et al., 2019; Liu et al., 2019) achieves significant success. However, it still requires a large amount of labeled data to reach optimal generalization of model. Thus, researchers gradually focus on semi-supervised text classification where only a few annotated data is provided. The success of semi-supervised methods results from the usage of abundant unlabeled data: Unlabeled documents in training dataset provide natural consistency regularization by constraining the model predictions to be invariant to small noises in text input(Xie et al., 2019; Miyato et al., 2016; Chen et al., 2020a). Despite mitigating the annotation burden, these methods are highly unstable in different runs and can still easily overfit on the very limited labeled data.

Inspired by the success of supervised contrastive learning under few-shot settings(Gunel et al., 2021; Chen et al., 2022), we hypothesize that the learned contrastive representation under this scenario can help us tackle aforementioned high variance issue and impose additional constraints to the model. Label information and feature structure can simultaneously be propagated from labeled examples to unlabeled ones. Thus, we devise a novel contrastive consistency schema to further boost model performance. To validate the effectiveness and robustness of FTCC, we conduct extensive experiments on four datasets. The result of our experiments confirms that FTCC can be leveraged to improve the performance of few-shot text classification.

Based on this motivation, the contributions of this paper are as follows:

- We integrate supervised contrastive learning objective into consistency-regularized semi-supervised framework to perform text classification under few-shot scenario.

- We devised a novel contrastive consistency schema to propagate feature structure from labeled data to unlabeled data dynamically.

- We demonstrated our model's superiority over state-of-the-art semi-supervised methods and analyze the contribution of each component of FTCC through ablation study and also visualize the learned instance representations, showing the necessity of each loss and the advantage of FTCC on representation learning over BERT fine-tuning with cross-entropy.

## 2 RELATED WORK

### 2.1 SEMI-SUPERVISED TEXT CLASSIFICATION

Increasing attentions are paid to semi-supervised text classification, since abundant unlabeled data are easier to obtain compared to labeled data. For instance, Miyato et al. (2016) adds perturbations to word embedding from bidirectional LSTM(Graves & Schmidhuber, 2005) and employs adversarial and virtual adversarial training in the text domain. Xie et al. (2019) enforces consistency regularization between unlabeled data and augmented data after back-translation. Chen et al. (2020a) introduces an interpolation-based augmentation and regularization technique to extend consistency training work and better leverage vast unlabeled data.

### 2.2 CONTRASTIVE LEARNING

Contrastive Learning recently achieves remarkable success in self-supervised representation learning on computer vision (He et al., 2019; Chen et al., 2020b) and natural language processing (Fang & Xie, 2020; Yan et al., 2021). The basic idea of unsupervised contrastive learning is that after generating different views of the sample example, the model adopts a loss function that can pull an anchor and a positive view closer and push the anchor away from other negative examples in the embedding space. Its effectiveness on downstream linear classification is justified through alignment and uniformity (Wang & Isola, 2020; Arora et al., 2019). Learning good generic representation is widely investigated. For example, in natural language field, Fang & Xie (2020); Wang et al. (2021) adopt data augmentations such as word substitution, back translation, word reordering to generate different views of samples. Gao et al. (2021b); Yan et al. (2021) adjust dropout ratio to generate positive pairs. Subsequently, researchers propose supervised contrastive learning(Khosla et al., 2020) loss that enforces representations of examples from the same class to be similar and the ones from different classes to be distinct. Gunel et al. (2021) adopts this loss to fune-tune pre-trained language model and brings astonishing improvement in low-resource scenario. Chen et al. (2022) introduces label-aware data augmentation and utilizes supervised contrastive learning to simultaneously obtain discriminative feature representations of input examples and corresponding classifiers in the same space.

## 3 METHODOLOGY

### 3.1 PROBLEM FORMULATION

The task of our model is to train a text classifier that efficiently utilizes both annotated data and unannotated data in semi-supervised learning. We denote that $D^l = \{(x_i^l, y_i), i = 1, \ldots, n\}$ as labeled training data and $D^u = \{x_i^u, i = 1, \ldots, m\}$ as unlabeled training data where $n$ is the number of the labeled examples, $m$ is the number of the unlabeled examples and $n \ll m$.

### 3.2 SUPERVISED CONTRASTIVE LEARNING

Unlike traditional fine-tuning pre-trained language model, we additionally includes a supervised contrastive learning term to fully leverage the supervised signals (Gunel et al., 2021; Khosla et al., 2020). This mechanism takes the samples from the same class as positive samples and the samples from different classes as negative samples. The contrastive loss and cross-entropy loss is defined as follows:

$$\mathcal{L}_{ce} = -\frac{1}{N} \sum_{i=1}^{N} CE(y_i \| p_\theta(y_i | x_i^l)) \tag{1}$$

$$\mathcal{L}_{scl} = -\sum_{i=1}^{N} \frac{1}{N_{y_i} - 1} \sum_{j=1}^{N} \mathbf{1}_{i \neq j} \mathbf{1}_{y_i = y_j} \log \frac{\exp(x_i^l \cdot x_j^l / \tau_{scl})}{\sum_{k=1}^{N} \mathbf{1}_{i \neq k} \exp(x_i^l \cdot x_k^l / \tau_{scl})} \tag{2}$$

where we work with a batch of training examples of size $N$, $x \in \mathbb{R}^d$ is $l_2$ normalized embedding on the [CLS] token from an encoder to represent an example. $N_{y_i}$ is the total number of examples in the batch that have the same label as $y_i$; $\tau_{scl} > 0$ is an adjustable scalar temperature; the $\cdot$ symbol

denotes the dot product; $y_i$ denotes the true label and $p_\theta(y_i|x_i^l)$ denotes the model classifier output for the probability of the $i_{th}$ example after softmax.

## 3.3 Consistency Training

Back-translation(Sugiyama & Yoshinaga, 2019; Edunov et al., 2018) is common data augmentation technique and can generate diverse paraphrases while preserving the semantics of the original sentences. We utilize back translations to paraphrase the unlabeled data to get augmented noisy data. Following Xie et al. (2019), we minimize the Kullback–Leibler divergence between the augmented view of the example and the original view of example to encourage their consistency and enforce the smoothness of the model. FTCC adopts the loss function as follows:

$$\mathcal{L}_{con} = D_{\mathrm{KL}}(p_{\tilde{\theta}}(y|x^u) \| p_\theta(y|a(x^u)))$$ (3)

where $a$ is back-translation transformation, $p_{\tilde{\theta}}(y|x^u)$ is an original example's label distribution through classifier, $p_\theta(y|a(x^u))$ is the back-translated example's label distribution through classifier. $\tilde{\theta}$ denotes the stop gradient of $\theta$, as suggested by VAT (Miyato et al., 2017).

## 3.4 Contrastive Consistency

Inspired by Xie et al. (2019); Wei et al. (2021), we conjecture that consistency regularization not only can propagate label information from label data to unlabeled data, but also can propagate feature structure in their latent space. Thus, we propose to encourage the consistency between the feature pattern of original examples and their corresponding augmented examples, which further refines deep clustering and classification accuracy on augmented data.

We randomly sample a batch of $N$ examples and perform back-translation on every example, resulting in $2N$ data points. Given one example and its corresponding back-translated example, we treat the other $2(N-1)$ examples as negatives. Then we denote the similarity between the example $x^u$ and the negatives $n_i(i \in \{1, ...., 2(N-1)\})$ in one batch as:

$$P(i) = \frac{exp(x^u \cdot n_i/\tau_{cc})}{\sum_{j=1}^{2(N-1)} exp(x^u \cdot n_j/\tau_{cc})}$$ (4)

Likewise, the similarity between the corresponding augmented example $a(x^u)$ and the negatives $n_i(i \in \{1, ...., 2(N-1)\})$ in one batch as:

$$Q(i) = \frac{exp(a(x^u) \cdot n_i/\tau_{cc})}{\sum_{j=1}^{2(N-1)} exp(a(x^u) \cdot n_j/\tau_{cc})}$$ (5)

where $\tau_{con}$ is a temperature hyperparameter different from $\tau_{scl}$ and every embedding is $l_2$ normalized.

Using KL Divergence as the measure of disagreement, we can impose the consistency between the probability distributions $P$ and $Q$ :

$$\mathcal{L}_{cc} = D_{\mathrm{KL}}(P \| Q)$$ (6)

## 3.5 Overall training objective

Above all, the overall training objective is:

$$\mathcal{L} = \mathcal{L}_{ce} + \lambda_1 \mathcal{L}_{scl} + \lambda_2 \mathcal{L}_{con} + \alpha \lambda_3 \mathcal{L}_{cc}$$ (7)

Motivated by Huang et al. (2022), we also dynamically release contrastive consistency signal. During the first half of training epochs, $\alpha$ is set to 0 to avoid false feature structure propagation. After half of the training epochs finish, the model becomes more stable on representing feature, and we gradually increase $\alpha$ as the epoch grows: $\alpha = \frac{2t-T}{T}$, where $T$ is the total number of epochs and $t$ is the current epoch number. Figure 1 shows an overview of our FTCC framework.

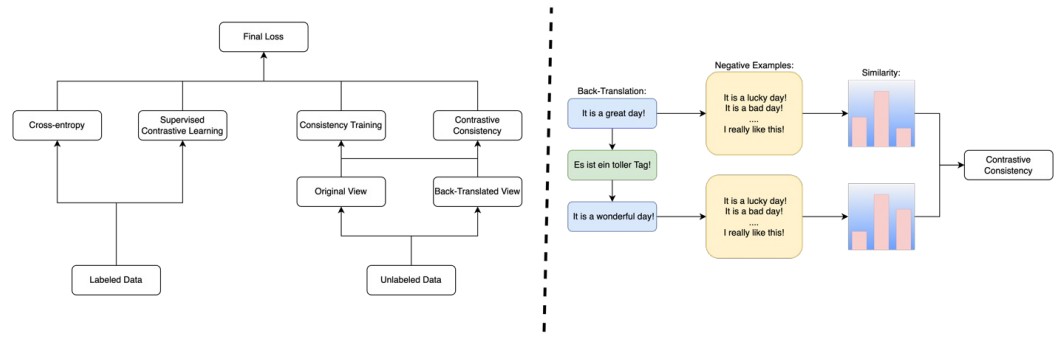

Figure 1: Left: The overall framework of FTCC. FTCC utilizes limited labeled data to compute cross-entropy and supervised contrastive learning loss and performs augmentation on unlabeled data to compute consistency training and contrastive consistency loss. Right: The illustration of contrastive consistency. FTCC computes the distances between the original or augmented views and the negatives in one batch to enforce their consistency and enhance the representation ability of the model.

## 4 EXPERIMENT

### 4.1 SET UP

We conduct our experiments on following four benchmark text classification datasets. SST2 (Socher et al., 2013) is a sentiment classification dataset of movie reviews. SUBJ (Pang & Lee, 2004) is a review dataset with sentences labelled as subjective or objective. PC (Ganapathibhotla & Liu, 2008) is a binary sentiment classification dataset that includes Pros and Cons data. IMDB (Maas et al., 2011) is a sentiment classification dataset from IMDB movie reviews. The dataset statistics are shown in Table 1. All datasets are in English language.

Table 1: Statistics of four text datasets. Unlabeled and Dev dataset have even label distribution.

|      | #Class | #Train | #Test | #Unlabeled | #Dev |
|------|--------|--------|-------|------------|------|
| SST2 | 2      | 7447   | 1821  | 3000       | 1000 |
| SUBJ | 2      | 9000   | 1000  | 3600       | 1000 |
| IMDB | 2      | 25000  | 25000 | 10000      | 4000 |
| PC   | 2      | 32097  | 13759 | 12000      | 4000 |

In few-shot learning scenario, we set the number of training labeled examples $K = 10$ per class. We follow Gao et al. (2021a) to randomly split training dataset into labeled data, unlabeled data and development data in 3 different samples. During splitting, we take label distribution and the size of dataset into account and do not use all unlabeled data. Three different models are trained on these splits. Then we evaluate the average performance of these three models on given test set.

We choose German as the intermediate language to perform data augmentation with Fairseq toolkit (Ott et al., 2019). We use pre-trained language model BERT as our backbone to encode all examples and use the input format "[CLS] sentence [SEP]" for all models. For three datasets SST2, SUBJ, PC, the max length are set to be the first 128, 128, and 64 tokens. In IMDB dataset, we remained the last 256 tokens, as suggested by Xie et al. (2019). We use Adam (Kingma & Ba, 2014) as the optimizer with linear decaying schedule. For simplicity, we set $\lambda_1$, $\lambda_2$, $\lambda_3$ to 1. For all datasets, the batch size of labeled data is 8 and the batch size of unlabeled data is 32, 24, 32, 32. Detailed information on experimental hyperparameter settings can be found in Table 2. We run experiments on one NVIDIA RTX A6000 GPU.

Table 2: Hyperparameter Configuration

|  | SST2 | IMDB | SUBJ | PC |
|---|---|---|---|---|
| max length | 128 | 256 | 128 | 64 |
| labeled batch size | 8 | 8 | 8 | 8 |
| unlabeled batch size | 32 | 24 | 32 | 32 |
| learning rate | 1e-5 | 2e-5 | 1e-5 | 1e-5 |
| max step | 2000 | 2500 | 2000 | 2500 |
| SCL temperature | 0.03 | 0.01 | 0.01 | 0.03 |
| CC temperature | 0.1 | 0.1 | 0.1 | 0.1 |
| warm-up percent | 0.1 | 0.1 | 0.1 | 0.1 |
| weight decay | 0.01 | 0.01 | 0.01 | 0.01 |

## 4.2 BASELINE

We compare our FTCC to five baselines: (1) BERT-FT: Fine-tuning the pre-trained BERT-based-uncased model on the labeled texts directly. (2) BERT-SCL (Gunel et al., 2021): Fine-tuning BERT with cross-entropy loss and supervised contrastive loss (3) DualCL (Chen et al., 2022): Dual contrastive learning framework learns feature representations of input examples and corresponding classifiers in the same space. It classifies documents based on their similarity with classifier representation (4) UDA (Xie et al., 2019): Unsupervised Data Augmentation uses limited labeled examples for supervised training and encourages model to have consistent prediction between unlabeled examples and corresponding augmented examples. (5) MixText (Chen et al., 2020a): MixText creates multiple augmented training examples by interpolating text in hidden space and then perform consistency training. To make fair comparison, we reproduce the results with the best hyperparameter configurations for all baselines.

Table 3: Performance (test accuracy(%)) comparison with baselines. We use 10 labeled examples for each class. The results are averaged after three random seeds to show the significance Dror et al.. ± denotes the standard error of the mean.

| K = 10 | SST2 | IMDB | SUBJ | PC |
|---|---|---|---|---|
| BERT-FT | 67.76±4.37 | 69.58±4.75 | 86.73±1.48 | 75.80±2.39 |
| BERT-SCL | 70.01±4.69 | 66.97±1.98 | 88.56±2.51 | 80.79±2.14 |
| DualCL | 77.32±2.82 | 68.89±0.25 | 83.03±1.36 | 82.75±0.94 |
| UDA | 73.43±12.05 | 89.06±1.50 | 92.70±0.70 | 88.54±3.65 |
| MixText | 75.10±10.56 | 83.38±3.31 | 92.83±0.40 | 89.21±1.73 |
| FTCC | **86.43±1.21** | **90.17±0.15** | **93.30±0.17** | **90.51±0.79** |

## 4.3 MAIN RESULT

The classification accuracy of all methods is shown in Table 3. FTCC outperforms all baselines by average. Compared to two basic fine-tune baselines, FTCC outperforms them by a large margin up to 20%. Compared to DualCL, FTCC does not use label-aware augmentation and instead uses back-translation to generate additional views of training examples. Both methods incorporate supervised contrastive learning while our method focuses on the usage of unlabeled data and achieves more competitive performance. Compared to UDA, FTCC discards its additional training techniques (Confidence-based masking, Sharpening Predictions Grandvalet & Bengio (2004), Training Signal Annealing) and uses supervised contrastive learning and contrastive consistency to prevent overfit and achieve better and more stable performance. Compared to MixText, FTCC does not redesign pre-train language model between different layers but adopts a simply but effective method to fine-tune it, which demonstrates that learning quality feature representation for data is more effective and efficient in low-resources scenario and FTCC is more robust and capable of fully leveraging labeled and unlabeled data.

## 4.4 VISUALIZATION

To explore how FTCC improves the quality of learning representations, we uses t-SNE (van der Maaten & Hinton, 2008) to visualize learned CLS embeddings on four test datasets. We shows the results in Figure 2. We can find that only using 10 examples per class to fine-tune BERT with cross-entropy loss has unstructured feature representation. However, our model can clearly separate examples from different classes and pull examples from same class closer. It demonstrates that the learned representations are more discriminative and enhance our model's robustness and generalization.

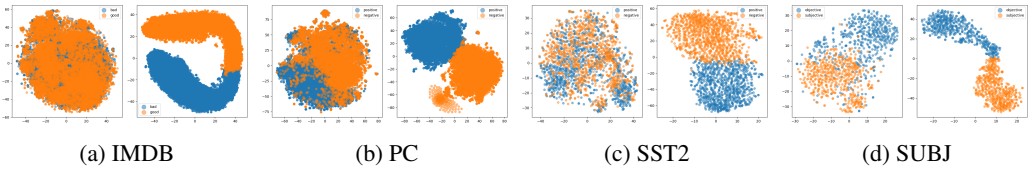

| (a) IMDB | (b) PC | (c) SST2 | (d) SUBJ |

Figure 2: t-SNE plots of learned sentence embeddings on four test datasets. Different colors denote different classes. In every pair, the left is cross-entropy, the right is FTCC.

## 4.5 ABLATION STUDY

To further examine the effectiveness of each component, we investigate FTCC with the following ablations by removing one component at a time: (1) do not perform supervised contrastive learning (**w/o. SCL**) (2) do not perform contrastive consistency (**w/o. CC**) (3) do not perform supervised contrastive learning (**w/o. CON**). The results are shown in Table 4. We can observe that every removal causes worse performance than original model, confirming the necessity of each loss term; Moreover, simply using **SCL** or **CON** has highly variance result while jointly incorporating them in our model design successfully alleviates this issue; **CC** component during the second half of training process further refines deep clustering and boosts FTCC's performance.

Table 4: Ablation study of FTCC. See abbreviation meanings in texts

| Ablations | SST2 | IMDB | SUBJ | PC |
|---|---|---|---|---|
| w/o. SCL | 73.51±9.26 | 87.07±1.16 | 90.13±2.50 | 86.64±4.37 |
| w/o. CC | 85.62±1.53 | 90.14±0.14 | 92.73±0.49 | 88.08±4.96 |
| w/o. CON | 78.56±2.25 | 84.88±5.73 | 92.40±0.17 | 88.84±3.12 |

## 5 CONCLUSION

In this paper, we propose FTCC that can effectively utilize unlabeled data under few-shot scenario and simultaneously enforce more compact clustering of sentence embeddings with similar semantic meanings. Moreover, we propose a contrastive consistency schema that can improve the quality of feature representation and the ability of model generalization. Our model achieves competitive performance on four text classification datasets, outperforming previous methods. Since using supervised contrastive learning to fine-tune pre-trained language model is a simple but effective training technique, in the future, we hope to incorporate this method into weakly-supervised or unsupervised NLP framework to maximally reduce human cost.

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
