# OpenReview forum: "Few-Shot Text Classification with Dual Contrastive Consistency Training"
_ICLR.cc/2023/Conference — Submitted to ICLR 2023_

### Official Review · Reviewer_CtgC · 2022-10-22

**Confidence:** 4
**Correctness:** 2
**Technical Novelty And Significance:** 1
**Empirical Novelty And Significance:** 1
**Recommendation:** 3

**Clarity, Quality, Novelty And Reproducibility:**

- The paper is already de-anonymized on paperswithcode.
- The approach of contrastive learning for few-shot classification is not new for text (https://doi.org/10.1609/aaai.v36i10.21292, arXiv:2205.01308) or other domains (https://arxiv.org/abs/2209.08224). It is not clear whether the specific addition of the consistency-regularized framework is enough of a novelty.

**Strength And Weaknesses:**

Strengths

- contrastive learning for few-shot learning in text is of course an interesting solution.
- the datasets chosen are standard for text-classification tasks, and the baselines are fine, generic approaches (although other baselines closer to the proposed method could have been selected, as per below)
- the empirical results (Table 3) are, of course, far better for the proposed method than the alternatives, across all selected datasets.

Weaknesses

- Few-shot learning in NLP is a very well explored problem and some key papers in this area, especially more recent ones than those cited (e.g., arxiv:2005.14165) are not considered.
- The benchmarking datasets of RAFT and ODIC should have been at least mentioned if not used.
- The ablation studies need to be better explained, within an overall better empirical framework.


**Summary Of The Paper:**


The paper considers few-shot text classification for pre-trained LMs. A supervised contrastive learning approach is taken, with consistency-regularization, and evaluation is performed across SST2, IMDB, SUBJ and PC datasets, against a collection of recent models, including BERT-SCL and DualCL.


**Summary Of The Review:**


Contrastive, few-shot learning in text is a popular area of research. This paper’s novelty is around consistency-regularization applied to this scenario, but this may not be sufficient, and overly simplistic empirical approaches should be updated with closer baselines, additional datasets, and more rigorous ablations.

---

### Official Review · Reviewer_6ZcP · 2022-10-23

**Confidence:** 4
**Correctness:** 2
**Technical Novelty And Significance:** 2
**Empirical Novelty And Significance:** 2
**Recommendation:** 3

**Clarity, Quality, Novelty And Reproducibility:**

The description of the algorithm and experiments are clear in this submission therefore I think the experiments can be reproduced given time. For novelty and quality, please see under Strength and Weaknesses.

**Strength And Weaknesses:**

My main concern with this submission is that it overlooks too much relevant work in this field. First of all, it is not true that all pre-trained language models are horrible few-shot learners. Actually some large pre-trained models, especially the auto-regressive ones, achieve quite surprising few-shot performance[1,2,3,4]. Second, a lot of existing work also shows that proper prompts[4,5] and proper fine-tuning methods[6,7] can also largely improve pre-trained models’ few-shot performance in general. Therefore, I think without discussing and comparing with such methods, the effectiveness or novelty of the proposed method are unclear.

My second concern is on the experiment side. As mentioned earlier, the method is only applied on BERT-base. More pre-trained models should be considered, especially those having good few-shot performances. Moreover, the number of shots, 10, is a bit random to me as well. Diverse numbers of shots need to be tested.  3 random seeds also sound too small for me because of the huge variance of few-shot experiments. Please also follow existing work to increase the number of random seeds[7].

[1] Brown, Tom, et al. "Language models are few-shot learners." Advances in neural information processing systems 33 (2020): 1877-1901.

[2] Raffel, Colin, et al. "Exploring the limits of transfer learning with a unified text-to-text transformer." J. Mach. Learn. Res. 21.140 (2020): 1-67.

[3] Radford, Alec, et al. "Language models are unsupervised multitask learners." OpenAI blog 1.8 (2019): 9.

[4]Gao, Tianyu, Adam Fisch, and Danqi Chen. "Making pre-trained language models better few-shot learners." arXiv preprint arXiv:2012.15723 (2020).

[5]Perez, Ethan, Douwe Kiela, and Kyunghyun Cho. "True few-shot learning with language models." Advances in Neural Information Processing Systems 34 (2021): 11054-11070.

[6]  Zhang, Ningyu, et al. "Differentiable prompt makes pre-trained language models better few-shot learners." arXiv preprint arXiv:2108.13161 (2021).

[7] Mahabadi, Rabeeh Karimi, et al. "Perfect: Prompt-free and efficient few-shot learning with language models." arXiv preprint arXiv:2204.01172 (2022).

**Summary Of The Paper:**

This paper first claims that in few-shot settings, fine-tuning pre-trained language models doesn’t achieve satisfactory performances. Therefore, this paper proposes a method to employ contrastive learning for few-shot fine-tuning of pre-trained models, due to the success of contrastive learning in other few-shot scenarios. The experiments are only conducted on pre-trained BERT-base model on 4 downstream NLP tasks.

**Summary Of The Review:**

please see under Strength and Weaknesses.

---

### Official Review · Reviewer_3uH4 · 2022-10-24

**Confidence:** 3
**Correctness:** 4
**Technical Novelty And Significance:** 2
**Empirical Novelty And Significance:** Not applicable
**Recommendation:** 3

**Clarity, Quality, Novelty And Reproducibility:**

Clarity: The paper is overall well-written and intelligible. The paper is however much shorter than the page limit, space that could have been used to good effect for more detailed background, explanations, further experiments and insights.

Quality: No major concerns with methodology.

Novelty: This is my main concern with this paper, see previous section.

Reproducibility: No major concerns. The proposed changes seem relatively easy to implement and experiments, including hyper-parameters, are detailed. The code was not open-sourced by its authors; since the contribution is mainly empirical, open-sourcing the code may benefit practitioners.

**Strength And Weaknesses:**

Overall, the paper provides interesting insights on simple ways to improve few-shot classification for text applications.

The novelty of this paper mostly lies in combination of various losses from the literature. Of the 3 additional loss terms used by the authors, two are applied as-is (modulo minor details) from existing papers and one is adapted from computer vision. Empirically, this combination is shown to allow improving classification accuracy and reducing variance.

While the paper is generally clear, the paper feels like a short paper. For an ICLR paper, I would have liked a deeper motivation and review section, as well as additional experiments and insights. For example:
- how do the results change when the number of labeled examples vary (e.g., 3, 10, 50, 100)?
- any theoretical/empirical reason to set all lambdas to 1?
- can the ablation study be better explained? I interpreted CC as the main contribution of the paper, but its impact seem minor relative to the both SCL and CON.

Finally, the exact differences, or lack thereof, between the losses in FTCC and other papers could at times be further clarified inside section 3.

**Summary Of The Paper:**

The paper proposes a novel Contrastive Consistency (CC) learning objective for few-shot text classification in semi-supervised settings (i.e., when additional unlabeled data are available). This new CC loss term is added when fine-tuning a BERT model, together with:
- a Supervised Constrastive Learning (SCL) loss (from Gunel et al, 2021)
- Consistency Training (CON) (adapted from Xie et al., 2019),
on top of the regular cross-entropy.

The authors show their method outperforms 4 recent baselines on 4 English datasets.




**Summary Of The Review:**

This paper is an easy and interesting read that could be useful to some ML practitioners. Its novelty seems however fairly limited and as such the paper feels short of the bar for ICLR. Since this paper contribution is mostly experimental, its experiment section could perhaps be more comprehensive and provide additional insights.

---

### Official Review · Reviewer_auWg · 2022-10-28

**Confidence:** 4
**Correctness:** 3
**Technical Novelty And Significance:** 2
**Empirical Novelty And Significance:** 2
**Recommendation:** 5

**Clarity, Quality, Novelty And Reproducibility:**

The clarify of the paper is okay for the technical details, but it should explain the 'FTCC' abbreviation the first time it appears. I still do not understand what the full model name is.

There are quality issues in the execution of the paper that has a negative impact on its significance. See the weaknesses section for details.

The novelty of the idea is good.

I have no reason to doubt the reproducibility of the idea. But I encourage the authors to release open source code whenever possible.

**Strength And Weaknesses:**

Strengths of the paper:
1) The idea of using back-translation for regularizing text classification models is a solid one.
2) The three proposes regularization constraints are reasonable.
3) Superior results compared to BERT and other baselines. In the literature, it is very hard to beat the BERT model for text classification tasks.
4) Good ablation study to demonstrate how each of the constraints are useful.

Weaknesses of the paper:
1) Only binary classification tasks where chosen. There are a large number of richer text classification datasets available in the literature.
2) The datasets used are very small. We do not know whether the model can still perform well when large number of training samples are available.
3) Traditional linear classifiers using bag-of-words, n-gram TFIDF etc is missing for the baselines. Along with BERT, these baselines are strong therefore should be considered in the study.
4) The ablation study should include a baseline classification result without any of the constraint, so that the readers can be clearer about the relative performance improvement.

**Summary Of The Paper:**

The paper proposes a text classification method that adds three regularization constraints. The first is a contrastive term between labeled samples depending on whether pairs of inputs have the same labels. The second is a consistency constraint using K-L divergency on back-translated transformation of the data samples. The third is a K-L divergence on the contrastive distribution between data samples and their back-translated transformations. The latter two can be performed on unlabeled data.

Experiments are performed on four binary classification tasks with number of training samples up to 32097 and number of unlabeled samples up to 12000. Superior results are observed compared to BERT and other data augmentation baselines. t-SNE analysis on the results are performed to show that the model obtains clear classification boundary. Ablation study is performed for each of the regularization contraints to show their respective necessity in improving the results.

**Summary Of The Review:**

Idea has good novelty. Execution needs improvement. Datasets for experiments are limited.

---

### Decision · Program_Chairs · 2023-01-20

**Decision:**

Reject

**Justification For Why Not Higher Score:**

All the reviewers voted reject and the authors gave up the rebuttal.

**Justification For Why Not Lower Score:**

All the reviewers voted reject and the authors gave up the rebuttal.

**Metareview: Summary, Strengths And Weaknesses:**

All the reviewers voted reject and the authors gave up the rebuttal.